# Study protocol for a non-randomised controlled trial: Community-based occupational therapy intervention on mental health for people with acquired brain injury (COT-MHABI)

**Marco Antonio Raya-Ruiz**[1,2], **María Rodríguez-Bailón**[3], **Beatriz Castaño-Monsalve**[4], **Laura Vidaña-Moya**[5], **Ana Judit Fernández-Solano**[6], **José Antonio Merchán-Baeza**[7] *

**1** Faculty of Health Science and Welfare, Social Sciences and Community Health Department, Universitat de Vic-Universitat Central de Catalunya (UVIC-UCC), Vic, Spain, **2** Specialised Support and Assessment Team (EASE), Institut Guttmann, Badalona, Spain, **3** Physiotherapy Department (Occupational Therapy). Universidad de Málaga, Málaga, Spain, **4** Neuropsychiatry, Institut Guttmann, Badalona, Spain, **5** Research Group GrEUIT., Escola Universitària d'Infermeria i Teràpia Ocupacional de Terrassa (EUIT), Universitat Autònoma de Barcelona, Terrassa, Spain, **6** Occupational Therapy Department, Universidad Católica San Antonio de Murcia, Murcia, Spain, **7** Faculty of Health Science and Welfare, Research Group on Methodology, Methods, Models and Outcomes of Health and Social Sciences (M3O), Universitat de Vic-Universitat Central de Catalunya (UVIC-UCC), Vic, Spain

* josan.merchan@uvic.cat

## Abstract

### Introduction

The sequelae of moderate-severe acquired brain injury (ABI) encompass motor, cognitive, sensory, emotional and behavioural areas that affect meaningful occupational participation and quality of life, with a high prevalence of associated mental disorders. When the patient returns to community life after discharge from the hospital, specialised care is generally insufficient due to the lack of consideration of the dual condition of mental disorder and ABI. Since there is a negative impact on competence and thus on occupational participation, occupational therapy represents a convenient way of intervention. On these assumptions, a community-based occupational therapy protocol on mental health for people with moderate/severe acquired brain injury (COT-MHABI) is presented. It is focused on meaningful occupational participation and looks for improvement in the quality of life.

### Methods and analysis

This study aims: (i) to design a protocol to evaluate the effectiveness of a community occupational therapy intervention based on MOHO for patients with a dual (mental health/ABI) for improving quality of life and self-perceived occupational performance; (ii) to analyse the outcomes of occupational and social variables (occupational balance, participation level, satisfaction with occupation and performed roles and community integration) after the COT-MHABI process; (iii) to analyse the impact of quality of life on satisfaction with occupations

**Data Availability Statement:** No datasets were generated or analysed during the current study. All relevant data from this study will be made available upon study completion.

**Funding:** JAMB has obtained the grant for Young Researchers of UVic-UCC. Code: AJI090221 Funder: Universitat de Vic-Universitat central de Catalunya (UVic-UCC). Funder URL: https://www.uvic.cat/en The funders had and will not have a role in study design, data collection and analysis, decision to publish, or preparation of the manuscript.

**Competing interests:** The authors have declared that no competing interests exist.

performed by this population. A non-randomised controlled clinical trial will be performed. Patients assigned to the experimental group will receive over one year of on-site and tele-matic occupational therapy sessions, 16 sessions on average. Variables such as quality of life, community integration or satisfaction with occupational performance will be collected at baseline, 6, and 12 months.

## Discussion

The needs for the dual mental/ABI population in their reintegration into the community are related to the associated deficits and to the absence of specialised services for the complexity of this patient profile. Few studies consider the coexistence of mental health and ABI issues. The COT-MHABI protocol is proposed to provide continuity to the community needs of this population, conceptualised from occupational participation, person-centred and focused on meaningful activities.

## Clinical trial registration

**Trial identifier and registry name** ClinicalTrials.gov ID: NCT04586842 https://clinicaltrials.gov/ct2/show/NCT04586842?term=252136&draw=2&rank=1; Pre-results; Community-based Occupational Therapy Intervention on Mental Health for People With Acquired Brain Injury (COT-MHABI).

## Introduction

Acquired Brain Injury (ABI) constitutes a major health problem throughout the world. It is a leading cause of mortality and a disability producer [1]. In Spain, it is prevalent in over 400,000 people [2]. ABI is defined as a brain injury after birth, resulting from an external force (as in the case of traumatic brain injury) or due to non-traumatic processes (stroke, anoxia, brain tumours, encephalitis, etc.) [3]. The sequelae of moderate-severe ABI encompasses motor, cognitive, sensory, emotional, and behavioural areas that affect occupational participation, social relationships, and the quality of life, often resulting in personality changes [4–12]. This population profile shows a significantly high prevalence of associated mental disorders either due to exacerbation of pre-existing symptoms, the onset of organic injury, or the resulting psychosocial situation [13–15].

Thus, people with moderate-severe ABI have, according to the World Health Organization (WHO), a highly complex condition, since it manifests deficits in bodily functions, limits activity, and restricts participation in life situations [16]. WHO itself recognises the importance of the concept of participation by defining it as the act of becoming involved in life situations [17]. Specifically, occupational participation encompasses the performance of meaningful activities that are part of the socio-cultural context. These are influenced by a person's performance capacity, habits and roles, volitional system (motivation for action), and environmental conditions [18]. Being able to participate in these valuable and meaningful activities has a direct benefit for well-being and life satisfaction [9,11,19].

Likewise, volition decreases, both in terms of control over one's own life (and, therefore, loss of active agency) [6,8] and in terms of one's motivation for action [20–22]. Occupational identity, i.e., a person's perception of himself or herself and who he or she wants to become, and occupational competence, the pattern of participation congruent with that identity [23],

are two concepts directly related to role performance [10,18,24], which is also significantly affected [6,25–27].

However, their occupational needs are not adequately covered in the community phase after hospital admission. There are deficiencies such as lack of information, guidance and specialised support, poor quality of care coordination, and scarce resources [2,28–30]. Furthermore, the confluence of previous or subsequent mental disorders interferes with the process of community reintegration, especially with regard to daily performance, because it is difficult to manage both by the person and by his or her social and family environment [13,26,31,32]. The specialised care they receive is generally insufficient because they are treated from a psychiatric or neurological perspective without considering the confluence of both diagnoses simultaneously in treatments and interventions, or they are directly rejected by these services [28,29,33,34], mainly due to lack of knowledge of the relationship between ABI and mental health and the invisibility of many of the associated deficits [31,35,36].

One of the most appropriate disciplines to support survivors in managing the consequences of the deficits produced by ABI is occupational therapy [10]. This is a profession committed to the promotion of health and well-being through occupation, identifying as its main objective the facilitation of people's participation in activities of daily living [10,37]. In other words, occupational therapy aims to help a person become what he/she wants to become through meaningful action framed within the daily routines and the roles played [18,38].

For this specific population, a person-centred intervention focused on the recovery of meaningful activities and occupations and accompanying the exploration of new and balanced participation options it is recommended [9,12,26]. More specifically, evidence has recommended the use of the Model of Human Occupation (MOHO) in mental health or ABI interventions [10,39–41]. This model, specific to occupational therapy, is the most widely used occupation-focused model in the world [10]. It provides specific assessments and interventions. It is based on four main aspects: motivation for occupation, habits and routines, occupational performance skills, and the influence of the environment on participation [10,23,24].

As shown above, there is evidence for the use of this occupational therapy model in both ABI and mental health populations, but there is a lack of studies and interventions referring to the dual situation of considering mental health issues in patients with ABI.

Due to this lack of evidence, the occupational needs discussed for this complex population and the absence of similar interventions, we present this design for a community-based occupational therapy protocol on mental health for people with moderate/severe acquired brain injury (COT-MHABI).

## Objectives

This study aims: (i) to design a protocol to evaluate the effectiveness of a community occupational therapy intervention based on MOHO for patients with a dual (mental health/ABI) for improving quality of life and self-perceived occupational performance; (ii) to analyse the outcomes of occupational and social variables (occupational balance, participation level, satisfaction with occupation and performed roles and community integration) after the COT-MHABI process; (iii) to analyse the impact of quality of life on satisfaction with occupations performed by this population.

## Materials and methods

### Inclusion and exclusion criteria

The study population (experimental and control group) will be composed of (i) adults aged between 18–50 with (ii) a diagnosis of medium or severe ABI and (iii) a diagnosis of mental

disorder following ABI (as recognised in the 5th edition of the Diagnostic and Statistical Manual of Mental Disorders (DSM-5) [42]) presenting behavioural, mood and/or emotional disorder. In addition, (iv) the patient must show difficulties in occupational participation compared with the pre-ABI situation and present occupational needs, as determined by an occupational therapist's comprehensive assessment plus the patient's identification of at least two occupational performance problems using the COPM (Canadian Occupational Performance Measure, a more detailed description of the tool in Data collection section of this paper). Patients also must be (v) in a situation of hospital discharge from ABI specialisation units and (vi) domiciled in the same province to which the providing hospital belongs.

Subjects will be excluded if (i) they are in a situation of symptomatological destabilisation with severe functional impairment that, as a priority, requires continued support from specialised mental health units and/or psychiatric or hospital admission, and/or (ii) they were diagnosed prior to the brain injury with a severe mental disorder (e.g., schizophrenia or a major depressive disorder) or dementia, and/or (iii) the cognitive deficits associated with ABI do not allow the agreed establishment of goals autonomously or with minimal support from the main caregiver.

## Recruitment methods

For both the experimental and control groups, the referral of patients to the study will take place through the neuropsychiatry department of the Institut Guttmann Neurorehabilitation Hospital in Badalona (Spain), which does not participate in the al-location of subjects, protocol intervention, data collection, or statistical analysis of the results. This department will assess mental health baseline status using the Neuropsychiatric Inventory Questionnaire (NPI-Q)) [43], evaluating possible interference with the results of concomitant pharmacological treatment.

Regarding the allocation, the experimental group will be composed of patients that will receive the COT-MHABI intervention protocol, framed within a community intervention programme at the hospital, after discharge. As it is a very specific and novel intervention, the resources for its implementation are limited, generating a waiting list (over 12 months). Given that this standby period is longer than the duration of the intervention in the experimental group, patients on the waiting list will be invited to participate in the control group with the consideration of receiving the intervention at a later stage.

## Trial design and setting

Due to the logistical reasons discussed above, a non-randomised controlled clinical trial will be performed. The measurement points of both groups will be comparable, matching gender, age, geographical area of residence and diagnosis. Fig 1 shows the SPIRIT 2013 schedule of enrollment, interventions, and assessments. Fig 2 includes a flow chart of the main parts of the study methods.

## Intervention

**Experimental group: Community-based occupational therapy intervention in mental health for people with ABI.** Participants in the experimental arm will receive the COT-M-HABI protocol, carried out after they are discharged and returned to their communities. This protocol is composed of two main phases. In the first phase, an occupational assessment will be carried out. Based on the results obtained in the assessment, goals setting will take place.

The second phase of the COT-MHABI protocol corresponds to the intervention process. This process will be composed of several sessions divided in stages implemented through an

| | STUDY PERIOD | | | | |
| --- | --- | --- | --- | --- | --- |
| | Enrolment | Allocation | Post-allocation | | Close-out |
| TIMEPOINT | *Baseline* | *Baseline* | *1st month* | *6 months* | *12 months* |
| **ENROLMENT:** | | | | | |
| **Eligibility screen** | X | | | | |
| **Informed consent** | X | | | | |
| **[Demographics, diagnosis]** | X | | | | |
| **Allocation** | | X | | | |
| **INTERVENTIONS:** | | | | | |
| *[Experimental group intervention]* | | | ●————————————● | | |
| *[Regular control group intervention]* | | | ●————————————● | | |
| **ASSESSMENTS:** | | | | | |
| *[Outcome variables]* | | | X | X | X |

**Fig 1. SPIRIT 2013 schedule.** SPIRIT schedule of enrollment, interventions and assessments timepoints along the study period (1 year maximum after the allocation).

occupation-based and occupation-focused intervention [44]. The structure of the intervention will be an adaptation from the MOHO Remotivation Process [18,23], which has as one of its main objectives the facilitation of occupational participation by people with severe volitional and motivational disorders [23]. The COT-MHABI protocol will follow the diagram shown in Fig 3.

Once a person agrees to participate in the study, an information sheet will be handed out, and the consent forms will be signed. The principal investigator, an occupational therapist with 15 years of experience, will carry out the intervention and the evaluations at the experimental group.

This intervention will have a duration of one year. During this time, a maximum of 12 on-site sessions and 4 synchronous telematic sessions in video-call format are foreseen. The on-site sessions, which will take place in a home and/or community environment, will last between 60 and 90 minutes, depending on the objectives set, and the telematic sessions will last approximately 45 minutes. In an exceptional case in which it is not possible to carry out the session in person or by video-call, it will be carried out by a telephone call. Likewise, both telephonic and asynchronous communication (e-mail or text messages) will be considered for possible exchanges of information or contingencies, without these communications constituting an intervention session per se.

During the COT-MHABI assessment phase, at the first session, an initial comprehensive assessment will be carried out, where the occupational therapist will collect information regarding the occupational history, i.e., meaningful activities for the person and their impact on the life narrative and about current occupational participation, i.e., the occupations that the person is effectively performing at the time of the assessment. During the second session, a specific assessment will be carried out, where a detailed analysis of relevant areas and facilitators/hinderers of occupational participation will be performed together with an exploration of significant roles and the person's expectations about them. Based on this information and the information gathered in the first session, the objectives will be established with the person's agreement.

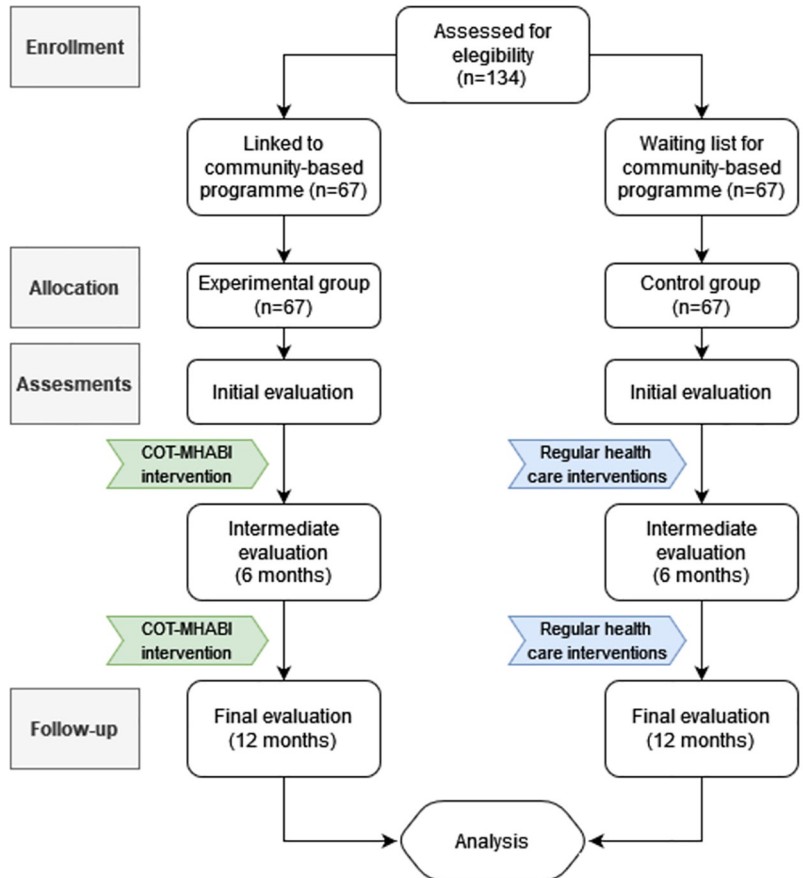

**Fig 2. Study methods flow chart.** 134 participants will be assessed for elegibility, allocating 67 in each arm (experimental and control groups). COT-MHABI protocol will be applied in experimental group and regular health care interventions in control group. For both groups, outcomes will be assessed over a one-year period at baseline, by the 6th and 12th month, when data collection is closed and analysis begins.

In order to achieve the objectives, the COT-MHABI intervention phase will be carried out during the rest of the process. This will be defined on the basis of three sequential and interrelated levels: (a) Occupational Exploration, (b) Occupational Competence, and (c) Occupational Achievement. Each level corresponds to a series of stages composed, in turn, of intervention strategies. A schematic summary of the composition of these levels can be seen in Table 1, while the complete list of strategies and goals corresponding to each level appears in the S1 Appendix *at Supporting Information* section of this protocol.

Thus, the general proposal of the intervention process will be to carry out a progressive path that enables the person to experience how to (i) explore new occupations through the development of awareness of one's own capacity in a safe environment, (ii) integrate these new learnings into habits congruent with the demands of the roles and the physical and social environment, experiencing an increase in personal effectiveness, and (iii) carry out an effective increase in occupational participation that has a permanent impact on occupational identity, in congruence with the pattern of life desired by the person. These strategies, proposals, and actions will be aimed at seeking personal autonomy through the use of meaningful activity, adaptation to the physical environment, and optimisation of the relationship with the social and family environment [18,23].

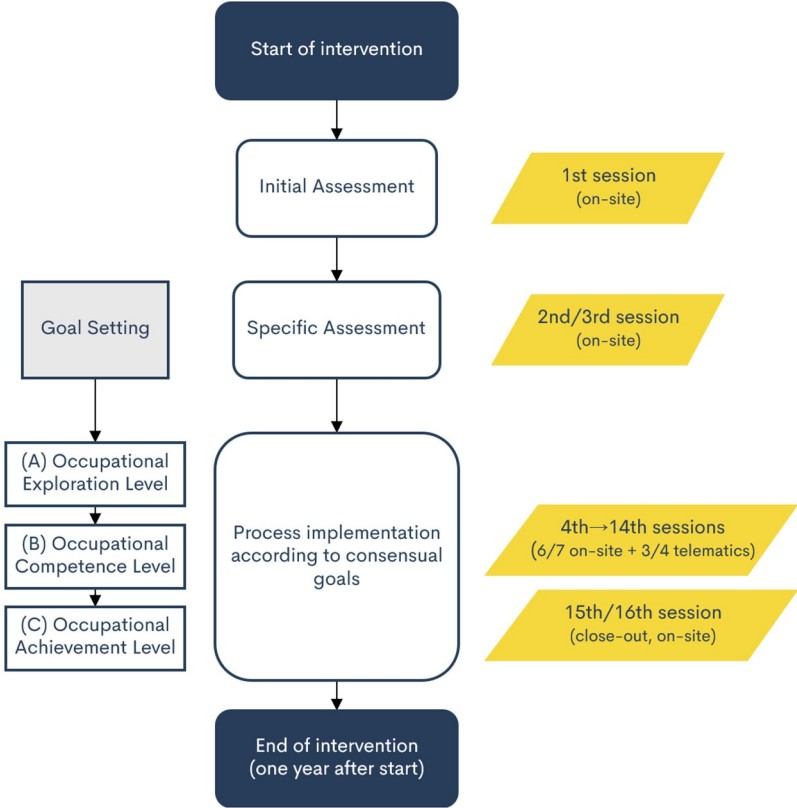

**Fig 3. Outline of the COT-MHABI intervention process.** The intervention will begin with an initial comprehensive assessment during the first session. The goals are agreed with the participant in the following sessions based on a specific assessment. Subsequently, between the 4th and the 16th session (on-site and telematics), the intervention will be implemented according to the goals and through the three-level process (Exploration, Competence and Achievement) based on the Model of Human Occupation.

The therapeutic intervention process will be complemented with counselling actions in relation to ABI and mental health for social and health care resources and, in general, for community agents involved in the planning of occupational participation of the person. Likewise, actions will be carried out to connect resources required by the intervention, always with the aim of facilitating an optimisation of occupational performance that fosters participation and occupational balance, both for the person and the family environment.

**Control group: Regular health care treatments (private/public health services).** People belonging to the control group will receive the usual services of their health centres, mainly access to outpatient consultations and, in some cases, private treatments. The duration of these treatments will be estimated by the centres themselves, depending on the patient's clinical condition and/or demand. A registration of the types of treatment and their duration will be carried out.

With the aim of improving control over the treatments received, those patients will be excluded from the study if they are receiving any MOHO intervention in the private or public service.

## Data collection

Outcome variables will be assessed in both study groups at the beginning of the COT-MHABI protocol, 6 months after the start of the protocol, and after the end of the protocol (12 months)

**Table 1. Composition of the intervention process levels.**

| Levels | Stages | Stage goals | Intervention strategy example |
|---|---|---|---|
| Occupational exploration | Validation | Enable access to initial experiences of capability through meaningful activity in a safe environment. | Accompaniment and support in initial experiences of capacity in acceptable and meaningful occupational forms. |
| | Willingness to explore | Favour an optimal basal state for allowing environmental exploration. | Provide strategies for promoting autonomy in managing difficulties related to mental well-being and post-ABI deficits. |
| | Election | Enable the person to increase his or her sense of capability during the exploration and choice of new habits and roles. | Facilitate exploration of new volitional opportunities according to the social and community environment. |
| | Effectiveness | Promote continued development of the person's sense of efficacy through exploration and preliminary participation in meaningful habits and roles. | Facilitate accessibility to occupational forms and tasks appropriate to the routines and chosen roles. |
| Occupational Competence | Internalisation of a sense of effectiveness | Foster self-analysis of capacity and effectiveness in performance, planning challenges and objectives, in congruence with aspects of habituation (habits and roles) and the physical and social environment. Promote the acquisition of management strategies to face the difficulties present in his/her occupational performance. | Provide counselling to the family and social environment to optimise the ability to detect and assess effectiveness milestones and their importance within the rehabilitation process. |
| | Occupational narrative building | Facilitate the realisation of occupational actions and roles that develop and improve affected skills (motor, processing, communication and interaction). Facilitate registration of the new occupational narrative of continuity (role project). | Facilitating the continuity of the process of inscribing a sense of effectiveness through positive feedback. |
| OccupationalAchievement | Achievement | Facilitate internalisation of habits and execution of meaningful roles and consolidation of occupational actions that improve occupational performance. Optimise the occupational balance of the person and family environment in relation to the new occupations achieved. Provide information and support for continued performance in the post-intervention phase. | Provide preventive strategies for barriers to achievement in occupational participation. |

using the scales detailed in Table 2. The collection will be carried out by different researchers. To minimise potential discrepancies, one researcher will allways evaluate the same participant.

**Main outcomes.** *Quality of life*. This will be measured by the WHOQoL-BREF (World Health Organization Quality of Life-BREF, Spanish version) [45,46]. This is a generic self-administered questionnaire created by the Quality of Life Study Group of the World Health Organization (WHO). The instrument has 26 questions, two general questions on quality of life and satisfaction with health status, and 24 questions grouped into four areas: Physical Health, Psychological Health, Social Relationships and Environment. Higher scores indicate better perceived quality of life. A 5-point Likert-type response scale is used. The WHOQoL--BREF shows excellent reliability (ICC = 0.87) for the quality of life variable in ABI due to traumatic brain injury [47].

*Perceived occupational performance and satisfaction with performance*. This will be evaluated by COPM (Canadian Occupational Performance Measure, Spanish version) [23,48], which is a self-administered occupational therapy measure based on client-centred practice that helps to establish occupational needs. It assesses changes in self-perceived occupational performance and satisfaction with occupational performance following intervention. The COPM shows excellent reliability (r = 0.87–0.88) in stroke patients [49].

**Secondary outcomes.** *Satisfaction with occupations and occupational balance*. This will be measured using the SDO-OB scale (Satisfaction with Daily Occupation and Occupational

**Table 2. Tests and variables used in the data analysis.**

| Test | Outcome variable definition | Scores and Domains | Reliability |
|---|---|---|---|
| *Main outcomes* | | | |
| WHOQoL-BREF | Quality of life | **Quality of life (Total score: Min 26-Max 130)** (Domains: general QoL, health status satisfaction, physical health, psychological health, social relationship and environment) | ICC = 0.87 |
| COPM | Self-perceived occupational performance | **Performance change**: difference between measures in self-perceived occupational performance **(Total score: Min 0-Max 50)** **Satisfaction change**: difference between measures in occupational performance satisfaction **(Total score: Min 0-Max 50)** | r = 0.87–0.88 |
| *Secondary Outcomes* | | | |
| SOD-EO | Satisfaction with daily occupation, occupational balance and activity level | **Satisfaction (Score: Min 13-Max 91)** Activity level (Score: Min 0-Max 13) Occupational Balance (Score: profile, no min or max) (Domains: productivity, leisure, home tasks and selfcare). | r = 0.84–0.92 |
| RCv3 | Satisfaction with performed roles | **Satisfaction (Score: Min 0.00-Max 1.00)** Number of performed roles (Score: Min 0-Max 10) Participation (Score: Min 0.00-Max 1.00) Desired Performance (Score: Min 0.00-Max 1.00) | r = 0.74–1.00 |
| ACS | Perceived activity participation level | **Percentage of maintained activities after ABI (Total score: Min 0%-Max 100%)** (Domains: instrumental activities, low physical demand leisure activities, high physical demand leisure activities, social and educational activities) | r = 0.88–0.95 |
| CIQ | Community integration and participation | **Community integration and participation (Total score: Min 0-Max 29)** (Domains: home integration, social integration and productivity) | r = 0.83–0.93 |
| FIM | Functional Independence | **Functional Independence (Total score: Min 18-Max 126)** (Domains: motor and cognitive) | ICC = 0.85 |

Balance, SDO-OB, Spanish version (SOD-EO)) [50,51]. The instrument assesses satisfaction in thirteen occupational areas, organised in four domains (productivity, leisure, housework and self-care) in terms of the person's activity level, occupational satisfaction and balance. Satisfaction level scores are answered based on a 7-item scale. The scale shows excellent reliability in mental health [52], both for performance satisfaction (r = 0.84) and activity level (r = 0.92). The version translated and validated in the Spanish context has shown good psychometric properties for the population with mental health disorders.

*Participation and satisfaction with the performed roles*. This will be evaluated by RCv3 (Role Checklist Version 3: Participation and Satisfaction, English version) [53]. The instrument assesses the person's perception of role performance, role satisfaction, and desire to participate in other roles in the future. This version has shown acceptable to excellent test-retest reliability for current role perception (κ = 0.74–1.00), desired future performance (κ = 0.77–0.98), and satisfaction with performance (α = 0.77–0.98) in the general population [54,55].

*Self-perceived level of participation in activities*. This will be measured using the ACS scale (Activity Card Sort, Spanish version) [56]. This is an instrument composed of a total of 89 photographs, used in occupational therapy to perform a joint exploration with the person, noting a catalogue of activities performed in all areas, both before and after the injury or situation that interferes with performance. It is useful to identify loss of participation, set goals and analyse evolution. The items include 20 instrumental activities, 17 social activities and 35 leisure activities of low physical demand and 17 of high demand. The original scale presents adequate psychometric characteristics, with an excellent level of test/retest reliability (r = 0.88–0.95) [56].

*Community integration*. This will be measured from the results collected in the CIQ (Community Integration Questionnaire, Spanish version) [57]. It assesses the limitations in the performance of social roles and community interaction of people with ABI. It presents three main

dimensions: domestic integration, social integration and productivity. The total score range is between 0–29 and most items have a range of 0–2. Higher values represent greater community integration and independence. The test-retest reliability of this scale is excellent for this population (r = 0.83–0.93) [57,58].

*Functional independence.* This will be valued from the FIM scale (Functional Independence Measure, Spanish version) [59,60]. The scale represents a uniform measurement system for disability based on the International Classification of Impairment, Disabilities and Handicaps. It assesses six functional areas (self-care, sphincter control, transfers, locomotion, communication and social cognition) within two domains (motor and cognitive). Each item is scored on a 7-point Likert-type scale. The items are performance-based rather than ability-based and are recorded in a hetero-administered manner. Reliability is excellent for the ABI population [60].

*Satisfaction with the intervention process.* This is a provider hospital's internal questionnaire (Spanish version). This will be collected qualitatively and quantitatively by regular programme satisfaction questionnaires. In this case, there is one questionnaire for the person participating in the process and another one for the main referring family member, if any. Both questionnaires are composed of seven questions about the subjective perception of the results obtained after the intervention with a 7-point Likert-type response scale. In addition, there are 4 more questions that, with a scale of 1 to 10, aim to collect the opinion of the quality of the intervention. Finally, a section will be included where, in a qualitative manner, the person can express other comments not included in the previous questions.

**Data management.** The data will be coded and stored by the research department of the patient provider, the Institut Guttmann Neurorehabilitation Hospital.

**Power analysis and sample size calculations.** Theoretically, with a two-tailed test, a sample of 134 participants will be needed in order to detect a significant difference in WHOQoL-BREF showing an effect size d = 0.49 (alpha = 0.05, beta = 0.20, power = 0.80), according to data collected in Chiu et al. (2016) [47]. The sample was calculated using the GPower 3.1 program.

## Statistical analysis

Data will be analysed on an intention-to-treat basis [61] i.e. results will be analysed using the basis of all participants. For missing data, the multiple imputation method will be used.

All variables will first be checked for normality. To test the effectiveness of the intervention, continuous variables having normal distribution will be analysed by repeated measures analysis of variance, ANOVA (within-subject factors: baseline, intermediate evaluation, final evaluation), comparing the two groups (between-group factors).

Due to the proposed design and the technical feasibility of the experimental group provider site, non-simultaneous recruitment makes the potential exposure to certain external factors unequal for some subjects. These factors may include the season of the year or the phase corresponding to the COVID-19 pandemic situation. To control them, we will explore the differences in the proportion of participants for these variables between the two groups using chi-square tests. If there are differences between groups an ANCOVA will be carry out with these external factors as covariates.

Non-normally distributed variables will be analysed using the Mann-Whitney test for group comparison. Friedman analyses and the Wilcoxon signed-rank test will be used for intrasubject factors. A p value < .05 will be defined as statistically significant. Bonferroni correction will be carried out for planned comparisons.

To determine the contribution of satisfaction with activities on quality of life in people with brain injury and a diagnosis of mental illness, beyond the intervention performed, a

hierarchical regression analysis will be performed with the entire sample, including the following variables in model 1: belonging to the intervention group, age and level of functionality. In model 2, in addition to the variables in model 1, the variable satisfaction with the occupation (collected by SDO-OB) will be added.

The quantitative analyses will be carried out using SPSS 25.0 (IBM Corporation, Armonk, NY). A qualitative thematic analysis of the responses obtained in the open-ended questions of the satisfaction questionnaires will be carried out. The analysis to be carried out will be inductive, and no pre-established categories will be used prior to coding. Three researchers will analyse the written fragments and will carry out an axial coding initially, and then carry out a selective coding, generating themes and categories. This analysis will be carried out using the Atlas.ti version 9 program.

## Ethics

The Good Clinical Practice (GCP) guidelines of the International Conference on Harmonisation (ICH) will be taken into account to carry out this study. This guarantees the protection of the rights, safety and well-being of the trial subjects in accordance with the principles of the Declaration of Helsinki, as well as the credibility of the data obtained. Likewise, the documentation provided to the people participating in the study complies with the Organic Law 3/2018, of December 5, on Personal Data Protection (LOPD) and guarantee of digital rights of the Spanish State. It also has the approval of the ethics committee of the neurorehabilitation hospital where the intervention takes place (Fundació Institut Guttmann, ID: 2019.306). In addtition, this study protocol has a trial identifier and registry name at ClinicalTrials.gov, ID: NCT04586842.

Written, informed consent will be obtained from all participants involved in the study. The ultimate authority to stop or modify the trial is the Institut Guttmann Neurorehabilitation Hospital. Also, the study is subject to reviews/audits of the process by the hospital's own research department, which is not actively involved in the design, collection or interventions.

## Limitations

Due to the technical feasibility of the programme, where the intervention is carried out, the clinical and social complexity of this patient profile, the lack of evidence on similar programmes and interventions, and the difficulties in finding a population in the setting that falls within the range of the inclusion and exclusion criteria, the presence of these issues that may interfere with the results and lead to bias should be highlighted.

The following have been identified as the main ones: (i) the inclusion of various aetiologies of ABI as a generalised concept, when the clinics presented by the patients may differ considerably [62]; (ii) the difficulty of knowing the influence of the different treatments that the patients in the control group may receive; (iii) the variability in the different conceptions of the construct "mental health" and its implications in the design of the protocol itself as well as in the practice on the quality of professional support that this population profile receives; (iv) due to the technical particularities of the research proposal, the therapist delivering the intervention and subsequently analysing a large part of the data is the same person, which may lead to biases in the results; (v) the heterogeneity in terms of the range of time after ABI in which the intervention takes place, as both groups have a wide range. This is because, at present, this is the first time that this type of support initiative is being provided in this region; so, patients are not screened according to the time of acquisition of the brain injury but by the importance of the current condition and their personal, occupational, social, and family issues.

## Discussion

The identified needs for the dual mental health/ABI population in their reintegration into the community are significantly related to the impairment of the deficits and their specific and major impact in daily life, causing a decrease in participation in meaningful activities, role performance, socialisation, and quality of life [10]. The identified factors of this deprivation are focused not only in the mentioned deficits but also on gaps in care such as limited availability of speciality services, financial strains, poor care coordination, and lack of tailored information, among others [3]. Regarding personal factors, an occupational identity loss appears concomitant with a devaluation of self [6].

To address these needs, the use of models and assessments such as those proposed in this study is supported by evidence. Several studies recommend the use of MOHO in mental health interventions, because of the improvement in volition, habituation, and skills [39–41], and in ABI interventions [10], due to the significant improvement in ADL performance, quality of life, and self-perceived health. Futhermore, the use of COPM and, in general, client-centred intervention oriented to occupational goal planning is recommended in community rehabilitation of ABI, even when volition is low or awareness of deficits interferes with daily performance [63], enabling the individual to feel more satisfied with his or her performance [64]. In addition, the fact that the setting of the COT-MHABI protocol is the home environment can induce a more client-centred approach, favouring the personalisation of the intervention by being in the everyday environment of the person [9].

Although there is evidence of the benefits of this kind of community-based interventions in ABI [65–68], there are few studies that develop the implications when both circumstances coexist [68,69]. This circumstance is crucial for providing adequate care. Also, focusing on occupational therapy intervention, one of the unmet needs for practitioners is the possibility of addressing the specificity of both conditions, that is, both those related to mental health and those involving deficits associated with ABI [69].

For this reason, it is necessary to propose research to assess the usefulness and effectiveness of community occupational therapy interventions like the COT-MHABI protocol that considers ABI and associated mental health issues. The aim is to add a further step in creating evidence for the problems of this ever-growing profile of patients [70,71] that go unnoticed, especially at their return to the community [2,3].

If the COT-MHABI protocol were to show significant results and show evidence in terms of improvement of the main and secondary variables, it would be a step forward in confirming that a community-based occupational therapy intervention, conceptualised from the facilitation of the development of occupational participation, increases the experience of meaning, wellbeing, and quality of life [6,10,11,23,26,27,64]. Given that the fundamental of intervention from occupational therapy is the recovery of meaningful activity performance, it implies a rethinking of the neurological patient's role, providing support in community reintegration as an individual, as family member, and as a citizen.

Being aware of the limitations, we propose this COT-MHABI study protocol as a basis for developing more targeted community-based occupational therapy intervention programmes in this type of patient's profile, as this is a growing population with better chances of survival and, as we have seen in the theoretical foundations outlined above, the continuity of social and health care is called into question, making them people at risk of exclusion.

We are confident that this type of intervention would have a direct impact on the individual, the family, and the referring health and social services. Considering variables, such as occupational balance or satisfaction with occupational performance, provides autonomous explorations and self-perception of competence to the person and, consequently, to their

family environment and carers. Finally, the dissemination of studies such as this one could provide more information for better management of the dual situations of mental health and acquired brain injury.

## Supporting information

**S1 Checklist. SPIRIT 2013 Checklist: Recommended items to address in a clinical trial protocol and related documents**\*.
(DOC)

**S1 Appendix.**
(PDF)

**S1 File.**
(PDF)

**S2 File.**
(PDF)

## Acknowledgments

The authors acknowledge the support received from Institut Guttmann and Universitat de Vic-Universitat Central de Catalunya (UVic-UCC).

## Author Contributions

**Conceptualization:** Marco Antonio Raya-Ruiz, María Rodríguez-Bailón, Beatriz Castaño-Monsalve, Laura Vidaña-Moya, José Antonio Merchán-Baeza.

**Funding acquisition:** José Antonio Merchán-Baeza.

**Investigation:** Marco Antonio Raya-Ruiz.

**Methodology:** Marco Antonio Raya-Ruiz, María Rodríguez-Bailón, José Antonio Merchán-Baeza.

**Project administration:** Beatriz Castaño-Monsalve, José Antonio Merchán-Baeza.

**Supervision:** María Rodríguez-Bailón, Beatriz Castaño-Monsalve, Laura Vidaña-Moya, Ana Judit Fernández-Solano, José Antonio Merchán-Baeza.

**Validation:** María Rodríguez-Bailón, José Antonio Merchán-Baeza.

**Visualization:** Marco Antonio Raya-Ruiz.

**Writing – original draft:** Marco Antonio Raya-Ruiz.

**Writing – review & editing:** Marco Antonio Raya-Ruiz, María Rodríguez-Bailón, Laura Vidaña-Moya, Ana Judit Fernández-Solano, José Antonio Merchán-Baeza.

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
