## [Decision Letter · Decision Letter 0]

10 Jun 2022

PONE-D-21-30716Community-based occupational therapy intervention on mental health for people with acquired brain injury (COT-MHABI): Study protocol for a non-randomised controlled trialPLOS ONE

Dear Dr. Merchán-Baeza,

Thank you for submitting your manuscript to PLOS ONE. After careful consideration, we feel that it has merit but does not fully meet PLOS ONE’s publication criteria as it currently stands. Therefore, we invite you to submit a revised version of the manuscript that addresses the points raised during the review process.

We look forward to receiving your revised manuscript.

Kind regards,

Vanessa Carels

Staff Editor

PLOS ONE

Journal Requirements:

Reviewers' comments:

Reviewer's Responses to Questions

**Comments to the Author**

1. Does the manuscript provide a valid rationale for the proposed study, with clearly identified and justified research questions?

Reviewer #1: Partly

2. Is the protocol technically sound and planned in a manner that will lead to a meaningful outcome and allow testing the stated hypotheses?

Reviewer #1: Partly

3. Is the methodology feasible and described in sufficient detail to allow the work to be replicable?

Reviewer #1: No

4. Have the authors described where all data underlying the findings will be made available when the study is complete?

Reviewer #1: Yes

5. Is the manuscript presented in an intelligible fashion and written in standard English?

Reviewer #1: Yes

6. Review Comments to the Author

You may also provide optional suggestions and comments to authors that they might find helpful in planning their study.

Reviewer #1: The manuscript entitled ‘Community-based occupational therapy intervention on mental health for people with acquired brain injury (COT-MHABI): Study protocol for non-randomized controlled

Trial’

The manuscript requires further improvement based on the following comments.

The title requires revision e.g. to begin with ‘Study protocol for a non-randomized controlled trial……..

Abstract

Page 2, the sentence ‘intervention based on MOHO and developed in a home setting’ requires revision. The comparative group is to be clearly stated. The abstract requires English editing,

Materials and methods

Page 6, typo etc.)).

Page 6, the numbering item A, B, C could be replaced with, (i), (ii), (iii) or (a), (b), (c). This applies to other sections of the manuscript.

Recruitment of subjects

Page 7, more information on how the subjects were allocated to intervention and control groups, who allocate/placed the subjects into the groups and collect the data to be clearly stated.

Page 7, how long is the estimated waiting list in the control group to be clearly stated.

Page 9, the objectives to be placed after introduction before materials and methods section. The objectives require English editing. ‘developed in a home setting’ not clear. The words 'meaningful occupation' and ‘ to know the contribution’ are to be revised. Meaningful occupation to be replaced with meaningful activities.

Page 8 Figure 1, more information/important point(s) on what was received by intervention and control group to be clearly stated in the figure.

Intervention

Page 11, the sentence ‘which has as one of its main objectives the facilitation of participation in occupations in people with severe volitional difficulties’ to be revised.

All inventories/questionnaires whether all are Spanish versions to be indicated.

Sample size calculation

One or two-tailed test and the attrition rate to be stated. Beta=0.20 and Power=1-0.20=0.80

Statistical analysis

A write-up on possible missing data and methods of handling missing data is to be added.

Page 20 Paragraph 2 Line 5, for the statement ‘In model 2, the variable satisfaction with the occupation (collected by SDO-OB) will be added’ to state clearly whether variables in model 1 are included in model 2 as well.

7. PLOS authors have the option to publish the peer review history of their article (what does this mean?). If published, this will include your full peer review and any attached files.

Reviewer #1: No

---

## [Author Response · Author response to Decision Letter 0]

17 Jun 2022

RESPONSE TO REVIEWERS - PONE-D-21-30716

Note: author uses red for responses under the original comment 

1. Does the manuscript provide a valid rationale for the proposed study, with clearly identified and justified research questions?

Reviewer #1: Partly

2. Is the protocol technically sound and planned in a manner that will lead to a meaningful outcome and allow testing the stated hypotheses?

Reviewer #1: Partly

3. Is the methodology feasible and described in sufficient detail to allow the work to be replicable?

Reviewer #1: No

Response to question 1, 2 and 3: Since some time has passed since our manuscript proposal and, after reading the reviewer's comments, we have revised most of the sections to bring them more in line with what was suggested. In this regard, we expect this sections to be considerably improved. Thank you for the comments. 

4. Have the authors described where all data underlying the findings will be made available when the study is complete?

Reviewer #1: Yes

5. Is the manuscript presented in an intelligible fashion and written in standard English?

Reviewer #1: Yes

6. Review Comments to the Author

You may also provide optional suggestions and comments to authors that they might find helpful in planning their study.

Reviewer #1: The manuscript entitled ‘Community-based occupational therapy intervention on mental health for people with acquired brain injury (COT-MHABI): Study protocol for non-randomized controlled Trial’

The manuscript requires further improvement based on the following comments.

The title requires revision e.g. to begin with ‘Study protocol for a non-randomized controlled trial……..

Response: Thank you for the suggestion. Change has been made 

Abstract

Page 2, the sentence ‘intervention based on MOHO and developed in a home setting’ requires revision. The comparative group is to be clearly stated. The abstract requires English editing. Response: Thank you for the suggestion. Changes have been made and, in general, an official translation service (PRS) has been used.

Materials and methods

Page 6, typo etc.)). Response: Fixed, thank you. 

Page 6, the numbering item A, B, C could be replaced with, (i), (ii), (iii) or (a), (b), (c). This applies to other sections of the manuscript. Response: Changes had been made plus best definition of inclusion/exclusion criteria

Recruitment of subjects

Page 7, more information on how the subjects were allocated to intervention and control groups, who allocate/placed the subjects into the groups and collect the data to be clearly stated. Page 7, how long is the estimated waiting list in the control group to be clearly stated. Response: Thanks for the comments, we have improved the definition of this section and we hope that it will further clarify the procedure.

Page 9, the objectives to be placed after introduction before materials and methods section. The objectives require English editing. ‘developed in a home setting’ not clear. The words 'meaningful occupation' and ‘ to know the contribution’ are to be revised. Meaningful occupation to be replaced with meaningful activities. Response: Thanks for the suggestion. We have changed the concept of home-setting to community-based, for being more complete. We have also corrected the phrasing of the objectives. Regarding the change from meaningful occupation to meaningful activity, we agree with your suggestion, also in line with articles such as Eakman (2013).

Page 8 Figure 1, more information/important point(s) on what was received by intervention and control group to be clearly stated in the figure. Response: on the basis of your suggestion, we have changed the figure to add this issue.

Intervention

Page 11, the sentence ‘which has as one of its main objectives the facilitation of participation in occupations in people with severe volitional difficulties’ to be revised. Response: revision done, thanks for the comment.

All inventories/questionnaires whether all are Spanish versions to be indicated. Response: changes have been made. 

Sample size calculation

One or two-tailed test and the attrition rate to be stated. Beta=0.20 and Power=1-0.20=0.80. Response: thank you very much for the comment, we have made changes in this aspect, adding the consideration of the two tails, also in the calculation of the power size.

Statistical analysis

A write-up on possible missing data and methods of handling missing data is to be added. Response: thank you so much for the suggestion. We have added a clarification in this respect, using the multiple imputation method.

Page 20 Paragraph 2 Line 5, for the statement ‘In model 2, the variable satisfaction with the occupation (collected by SDO-OB) will be added’ to state clearly whether variables in model 1 are included in model 2 as well. Response: thanks for the comment, we have better clarified that sentence.

---

## [Decision Letter · Decision Letter 1]

12 Jul 2022

PONE-D-21-30716R1Study protocol for a non-randomised controlled trial: Community-based occupational therapy intervention on mental health for people with acquired brain injury (COT-MHABI)PLOS ONE

Dear Dr. Merchán-Baeza,

Thank you for submitting your manuscript to PLOS ONE. After careful consideration, we feel that it has merit but does not fully meet PLOS ONE’s publication criteria as it currently stands. Therefore, we invite you to submit a revised version of the manuscript that addresses the points raised during the review process.

Please address the points that the reviewer noted and see the "additional Editor comments" section below.

We look forward to receiving your revised manuscript.

Kind regards,

Thomas Tischer

Staff Editor

PLOS ONE

Additional Editor Comments (if provided):

We noticed that specifically the abstract needs some copy editing. PLOS ONE does not provide copy editing services, please ensure the use proper English language and grammar throughout the manuscript.We noticed that the sample size of your study has increased from 106 to 134. This does not agree with the registered trial protocol NCT04586842. Please comment on this and seek out advise if an amendment is required. If an amendment is required, please ensure it is granted before the manuscript is resubmitted. If no amendment is required please explain why.

Reviewers' comments:

Reviewer's Responses to Questions

**Comments to the Author**

1. Does the manuscript provide a valid rationale for the proposed study, with clearly identified and justified research questions?

Reviewer #1: Yes

2. Is the protocol technically sound and planned in a manner that will lead to a meaningful outcome and allow testing the stated hypotheses?

Reviewer #1: Partly

3. Is the methodology feasible and described in sufficient detail to allow the work to be replicable?

Reviewer #1: Yes

4. Have the authors described where all data underlying the findings will be made available when the study is complete?

Reviewer #1: Yes

5. Is the manuscript presented in an intelligible fashion and written in standard English?

Reviewer #1: Yes

6. Review Comments to the Author

You may also provide optional suggestions and comments to authors that they might find helpful in planning their study.

Reviewer #1: The authors have put in a great effort to address the comments.

Minor comments

Line 358 - Typo error 'anaylisis'

Line 128-133 - Objectives to be placed at Line 127 (before Materials and Methods section)

7. PLOS authors have the option to publish the peer review history of their article (what does this mean?). If published, this will include your full peer review and any attached files.

Reviewer #1: No

---

## [Author Response · Author response to Decision Letter 1]

27 Jul 2022

Thank you so much for your feedback. We believe that this project is useful for the population profile for which it is designed, so we are enthusiastic with the opportunity to publish it

---

## [Editor Report · Decision Letter 2]

24 Aug 2022

Study protocol for a non-randomised controlled trial: Community-based occupational therapy intervention on mental health for people with acquired brain injury (COT-MHABI)

PONE-D-21-30716R2

Dear Dr. Merchán-Baeza,

We’re pleased to inform you that your manuscript has been judged scientifically suitable for publication and will be formally accepted for publication once it meets all outstanding technical requirements.

Kind regards,

George Vousden

Staff Editor

PLOS ONE
---

## [Editor Report · Acceptance letter]

29 Sep 2022

PONE-D-21-30716R2 

Study protocol for a non-randomised controlled trial: Community-based occupational therapy intervention on mental health for people with acquired brain injury (COT-MHABI) 

Dear Dr. Merchán-Baeza:

I'm pleased to inform you that your manuscript has been deemed suitable for publication in PLOS ONE. Congratulations! Your manuscript is now with our production department. 

Kind regards, 

on behalf of

Dr. George Vousden 

Staff Editor

PLOS ONE